# CLCA2: A Potential Guardian against Premature Senescence and Skin Aging

**DOI:** 10.3390/biomedicines12030592

**Published:** 2024-03-06

**Authors:** Lena Guerrero-Navarro, Ines Martic, Christian Ploner, Pidder Jansen-Dürr, Maria Cavinato

**Affiliations:** 1Institute for Biomedical Aging Research, Universität Innsbruck, 6020 Innsbruck, Austria; lena.guerrero-navarro@uibk.ac.at (L.G.-N.); ines.martic@uibk.ac.at (I.M.); 2Center for Molecular Biosciences Innsbruck (CMBI), 6020 Innsbruck, Austria; 3Department of Plastic, Reconstructive and Aesthetic Surgery, Medical University of Innsbruck, 6020 Innsbruck, Austria; christian.ploner@tirol-kliniken.at

**Keywords:** CLCA2, skin aging, senescence, UVB, tBHP, Nutlin3a

## Abstract

Cellular senescence, a state of irreversible growth arrest, is implicated in various age-related pathologies, including skin aging. In this study, we investigated the role of CLCA2, a calcium-activated chloride channel accessory protein, in cellular senescence and its implications for skin aging. Utilizing UVB and Nutlin3a-induced senescence models, we observed the upregulation of CLCA2 at both transcriptomic and proteomic levels, suggesting its involvement in senescence pathways. Further analysis revealed that the depletion of CLCA2 led to accelerated senescence onset, characterized by classic senescence markers and a unique secretome profile. In 3D skin equivalent models, SEs constructed with CLCA2 knockdown fibroblasts exhibited features reminiscent of aged skin, underscoring the importance of CLCA2 in maintaining skin homeostasis. Our findings highlight CLCA2 as a novel regulator of cellular senescence and its potential implications for skin aging mechanisms.

## 1. Introduction

The aging process is often associated with tissue impairment, which leads to the loss of tissue homeostasis [1]. This decline in tissue function has been correlated with the accumulation of senescent cells, which contribute to the onset of age-related diseases [1].

Cellular senescence is a multifaceted state in which cells cease to divide and undergo various changes, such as alterations in gene expression and morphology. It is often triggered by different factors, among them being telomere attrition [2], which results from a high number of cell divisions and induces the process of replicative senescence [3]. Additionally, senescence can be prematurely induced by external damaging stimuli such as different types of ionizing and ultraviolet radiation that cause damage to the DNA and cellular organelles [4]. This type of senescence is referred to as stress-induced premature senescence (SIPS).

While the appearance and subsequent elimination of senescent cells are crucial for wound healing and organismal development [5,6], during aging, these cells fail to be removed [7]. The persistence of senescent cells in diverse tissues promotes inflammation, primarily through the activity of the senescence-associated secretory phenotype (SASP), which contributes to the deleterious remodeling of the tissues [7].

Independent of the senescence-inducing stimuli, a persistent DNA damage response (DDR) and the activation of pathways such as p53/p21 and p16/pRB, which lead to permanent cell cycle arrest, are common characteristics of senescent cells [8]. During this process, p53 is stabilized by post-translational changes, which promotes the expression of CDK inhibitors such as p21 and p16. As a result, pRb is dephosphorylated and activated, and thus, the activity of the E2F transcription factor family is repressed, and the cell cycle is arrested [8].

Senescent cells exhibit considerable heterogeneity, posing a challenge to researchers who study senescence. This diversity arises from the broad spectrum of factors that can induce a senescent phenotype and the complex secretome displayed by these cells. The SASP encloses different types of cytokines, chemokines, matrix metalloproteases, growth factors and other secreted molecules, reflecting the metabolic flexibility of senescent cells and the various pathways activated in response to specific types of damage. Notably, Mitochondrial Dysfunction-Associated Senescence (MiDAS) features a distinct SASP characterized by the absence of the IL-1 inflammatory arm, while other types of senescence induce a strong IL-1 response [9]. Therefore, there is no single unequivocal method or universal marker to identify senescent cells. Typically, researchers employ a combination of several techniques to substantiate the identification of these cells [10].

In the context of skin aging, the persistence of senescent cells is recognized as a significant contributor to the aging process. Senescent cells accumulate within the skin over time, disrupting its homeostasis and functionality. This accumulation is associated with various age-related alterations, including impaired wound healing, diminished regenerative capacity, and compromised barrier function. The pro-inflammatory SASP of senescent cells plays a pivotal role in intercellular communication within the skin microenvironment. Factors secreted by senescent cells as part of the SASP can act on different skin compartments, contributing to the degeneration observed during the aging of this tissue. This dysregulated communication network exacerbates tissue inflammation, compromises cellular integrity, and accelerates the aging phenotype [11].

In this study, we have explored the role of chloride channel accessory 2 (CLCA2), a member of the CLCA family [12], in different types of cellular senescence. This family of proteins is known to regulate chloride channels in a calcium-dependent manner [12]. CLCA2 can be induced by DNA-damaging agents and is dependent on p53 regulation [13,14]. Additionally, CLCA2 expression increases during replicative senescence, where it takes part in senescence induction downstream p53 signaling [15]. Despite these associations, the specific function and mechanisms by which CLCA2 influences senescent cells remain unclear.

Our results suggest that CLCA2 is a key player in the processes of the UVB- and Nutlin3a-induced senescence of human dermal fibroblasts (HDFs), highlighting its significance in certain forms of SIPS. Importantly, CLCA2 appears to play a protective role for HDFs, as its knockdown accelerates the manifestation of senescent characteristics. Furthermore, these findings align with our studies conducted using 3D skin models, which demonstrate that the presence of CLCA2 in dermal fibroblasts supports appropriate epidermal differentiation, while the absence of CLCA2 in the dermis induces age-related characteristics in these models.

## 2. Materials and Methods

### 2.1. Cell Culture

Human dermal fibroblasts, HFF-2, were obtained from ATCC (Manassas, VA, USA, SCRC-1042). Cells were cultured in low-glucose DMEM media (Sigma, Darmstadt, Germany, D5921) supplemented with 10% heat-inactivated fetal bovine serum (FBS) (Gibco, Thermo Fisher Scientific, Waltham, MA, USA,10270-106), glutamine 4 mM (Sigma, Darmstadt, Germany, G7513) and penicillin/streptomycin 1% (Sigma, Darmstadt, Germany, P4333) at 37 °C, 5% CO_2_ in a humid incubator.

Cell numbers were counted using a CASY counter (OMNI Life Science, Bremen, Germany) after being washed twice with PBS and detached with trypsin (Sigma, Darmstadt, Germany, T3924). The cPDLs were calculated using the equation PDL = (lnF − lnI)/ln2, with F being the number of cells at the end of one passage and I the number of cells that were seeded at the beginning of the passage.

For UVB treatment [16], irradiated fibroblasts were seeded in 10 cm plates at a density of 6 × 10^5^ and control fibroblasts were seeded at 3 × 10^5^. Cells were washed twice with Hank’s balanced salt solution (HBSS, Sigma, Darmstadt, Germany, H8264) and covered with 2.5 mL of HBSS. Cells were irradiated with 0.05 J/cm^2^ in a Bio-Sun System (Vilver Lourmat) twice a day for 4 consecutive days. Protein and RNA lysates were collected on day 4 and 9.

For tert-butyl hydroperoxide (tBHP) treatment, treated fibroblasts were seeded in 10 cm plates at a density of 5 × 10^5^ and control fibroblasts were seeded at 3 × 10^5^. Cells were incubated in DMEM media containing tBHP (40 µM) for 1 h twice a day for 4 consecutive days as described [17]. Protein and RNA lysates were collected on day 4 and 9.

Nutlin3a (Sigma, Darmstadt, Germany, SML0580) treatment was applied following the protocol described in [18]. Briefly, cells were seeded at a density of 5 × 10^5^ in 10 cm plates and cultured for one week in medium containing 5 µM of Nutlin3a. On day 7, the cells were counted and replated in Nutlin3a-free medium. Analysis was conducted on day 8.

### 2.2. Immunoblotting

Cells were washed twice with cold PBS (Pan-Biotech, Aidenbach, Germany, P04-36500) and scraped, and protein was extracted using radioimmunoprecipitation assay buffer (RIPA) supplemented with protease inhibitors: 50 mM NaF, 2 μg/mL Aprotinin, 1 mM PMSF, and 1 mM activated Na-Orthovanadate.

Protein lysate concentrations were measured using the BCA protein assay kit (Pierce BCA Protein Assay Kit, Thermo Fisher Scientific, Waltham, MA, USA, 23225), and 20 µg of protein was separated by SDS-PAGE electrophoresis. Proteins were transferred to PVDF membrane (Immun-Blot, Bio-Rad, Feldkirchen, Germany, 1620177) using 1X transfer buffer (containing 10% methanol) in a wet chamber at 4 °C with a voltage fixed at 100 V for 1 h and 10 min. Membranes were blocked in 5% non-fat milk (Sigma, Darmstadt, Germany, 70166) in TBS-T (Tween 20 0.01%) for 1 h at RT and incubated overnight with the corresponding primary antibodies: CLCA2 (#HPA047192, Sigma Darmstadt, Germany), Phospho-Rb Ser807/811 (#9308, Cell signaling, Danvers, MA, USA), Lamin B1 (#ab16048, Abcam, Waltham, MA, USA), and GAPDH (#0411:sc-47724, Santa Cruz, Dallas, TX, USA). Incubation with the secondary antibody conjugated to HRP was performed in 1 h at RT. Membranes were exposed using the HRP substrate (Immonilon Western, Millipore, Darmstadt, Germany, WBKLS0500) at the Chemidoc Imaging System (Bio-Rad). Densitometric measurements were conducted utilizing ImageJ software (version 1.54b, National Institutes of Health, Bethesda, Maryland, USA) with GAPDH serving as the normalization standard.

### 2.3. Quantitative Real-Time PCR

RNA was extracted using the RNeasy Mini kit (Qiagen, Hilden, Germany, 74004), and cDNA synthesis was performed using the High-Capacity cDNA Reverse Transcription Kit (Thermo Fisher Scientific, Waltham, MA, USA, 4368814) following the manufacturer’s guidelines. Real-time PCR was performed in triplicate for each sample with AceQ^®^ Universal SYBR^®^ Green qPCR Master Mix (Vazyme, Nanjing, China, Q511-02) in the QuantStudio™ 7 Flex Real-Time PCR System (Thermo Fisher Scientific, Waltham, MA, USA, 4485701). Primers were purchased from Eurofins genomics and were the specific primer sequences from 5′-3′: CLCA2 (FW: TTC CTG GGA GCT GGA GTA CA, RE: GCT TTC CAT GTG GCA GGT AT), NEFL (FW: AAG ACC CTG GAA ATC GAA GC, RE: TCG TGC CAT TTC ACT CTT TG), SCG5 (FW: CTC ACC AGG CCA TGA ATC TT, RE: GTC TGG GTA CCC CTG ATC CT), SERPINB2 (FW: ATG GTC TAC ATG GGC TCC AG, RE: TGC AAA ATC GCA TCA GGA TA), WDR63 (FW: CCT GGA AAT GAG CTT CTG CT, RE: TTT CAC TGC CCA AAG AAA CC), B2M (FW: GAA TTC ACC CCC ACT GAA AA, RE: CTC CAT GAT GCT GCT TAC A). The housekeeping gene B2M was used for normalization.

### 2.4. Analysis of CLCA2 Membrane Expression by FACS

For the flow cytometry analysis, fibroblasts were first detached from culture plates using trypsin and subsequently counted. A total of 3 × 10^5^ cells were aliquoted into each tube. The cells were then stained with a CLCA2 antibody (#WH0009635M1, Sigma Darmstadt, Germany) at a dilution of 1:100 and incubated for 30 min at 4 °C. Following this, the cells underwent a washing step with PBS. Subsequently, they were incubated with an Alexa Fluor 488-conjugated anti-mouse secondary antibody at a dilution of 1:250 for 30 min at 4 °C. After a final wash with PBS, the cells were analyzed using a FACS Canto II flow cytometer. The fluorescence intensity at a wavelength of 488 nm was specifically measured to assess the binding of the CLCA2 antibody.

### 2.5. Stable Knockdown of CLCA2

After expanding the bacterial culture, plasmid DNA from SMARTvectorTM lentiviral shRNA glycerol stocks (Dharmacon, Lafayette, CO, USA) was extracted using the Qiagen^®^ Plasmid Maxi Kit (Hilden, Germany, 12165) in accordance with the manufacturer’s instructions. Lentiviral particles were produced as described [16]. The infection of both HDFs and HSDFs was performed using supernatant containing viral particles at MOI of 1. Starting two days after infection, cells were cultured with puromycin to select transduced cells. A scrambled shRNA lentiviral vector served as the control.

### 2.6. SA-β-gal Cytochemistry Assay

Cells were grown on 6-well plates and fixed for 5 min with 2% formaldehyde and 0.4% glutaraldehyde in PBS after washing 3 times with PBS. Cells were washed with PBS and covered with staining solution with a pH of 6.0 (150 mM NaCl, 2 mM MgCl, 5 mM potassium ferricyanide, 5 mM potassium ferrocyanide, 40 mM citric acid, 12 mM sodium phosphate, and 1 mg/mL 5-bromo-4-chloro-3-indolyl-b-D-galactoside (X-gal)) and incubated 24 h at 37 °C without CO_2_. The reaction was stopped by washing 3 times with PBS. SA-β-gal positive cells were detected by light microscopy with a Nikon eclipse TE300.

### 2.7. Cytokine Arrays

Cytokine arrays (Abcam, Waltham, MA, USA, ab133998) were performed with supernatants obtained from CLCA2KD and SCR HDFs according to the manufacturer’s protocol. The procedure involved incubating the cytokine array membranes with the samples, followed by a series of washes to prepare the membranes for further incubation with both Biotin-Conjugated Anti-Cytokines and HRP-Conjugated Streptavidin to allow the chemiluminescence detection. These assays were conducted using biological duplicates for each sample condition. For each cytokine, fold changes were calculated based on densitometric data comparing CLCA2KD and SCR cells across different passages. Cytokines displaying an absolute fold change greater than 2 in at least one passage were selected as significant. Z-score normalization was then applied to individual cytokines across all conditions for the heatmap representation.

### 2.8. Extracellular Vesicle (EV) Isolation and Characterization

Serum-depleted medium was applied to the cells, and conditioned medium was collected after 72 h. Conditioned media were centrifuged at 700× *g* for 5 min at 4 °C to exclude cellular debris, followed by centrifugation at 2000× *g* for 10 min at 4 °C to exclude apoptotic bodies and larger particles. The resulting supernatant was further centrifuged at 100,000× *g* for 90 min. The resulting pellets consisting of EVs were resuspended in 0.22 µm-filtered phosphate-buffered saline (PBS). To determine the size and concentration of extracellular vesicles, we utilized the nanoparticle tracking analysis from the Zetaview system (Particle Metrix, Ammersee, Germany). The machine was calibrated using 100 nm polystyrene standard beads (Particle Metrix, Ammersee, Germany). Settings were adjusted to the highest and lowest concentration of samples, and 11 different positions of the camera field were measured at least three times. To characterize EVs surface markers, 3.4 × 10^8^ total particles were incubated with CD81 magnetic beads (Invitrogen, Thermo Fisher Scientific, Waltham, MA, USA) overnight on a shaking plate. The CD81-captured particles were stained using appropriate antibodies to detect the tetraspanins CD9, CD63, and CD81 (Miltenyi Biotech, Bergisch-Gladbach, Germany, #130-113-438, #130-118-076, #130-118-342) and analyzed by LSRFortessa (BD Biosciences, San Jose, CA, USA) and Imagestream XMK II (Merck, Darmstadt, Germany) flow cytometry. Pure magnetic beads and antibody isotypes were used for gating and as negative controls, respectively. FlowJo software version 10.9.0 (BD Biosciences, San Jose, CA, USA) was used for the analysis of the obtained data.

### 2.9. Skin Equivalent (SE) Production

To produce 3D SEs, we employed either a wild type (WT) subjected to 4 days of UVB irradiation or a non-irradiated scrambled control (SCR) and CLCA2 knockdown (CLCA2KD) human skin dermal fibroblasts (HSDFs), along with matched human skin epidermal keratinocytes (HSEKs) from the same patient, a healthy 35-year-old male. The SEs were produced in technical triplicates to ensure the reproducibility and reliability of the results. The method for SE production followed the protocol described in [19]. In compliance with ethical committee standards, the HSDFs and HSEKs were isolated from donated skin provided by the Department of Plastic, Reconstructive, and Aesthetic Surgery at the Medical University of Innsbruck.

### 2.10. Hematoxylin–Eosin Staining

SEs underwent fixation in 4% paraformaldehyde overnight, followed by dehydration, clearing, and embedding in paraffin. Sections with 5 µm thickness were prepared from the paraffin-embedded skin equivalents (SEs). These sections were stained by regular hematoxylin–eosin and mounted in Entellan. The prepared slides were then examined under a Leica DMLS type 020-518.500 optical microscope for histological analysis. The epidermal thickness was measured with Image J Software (version 1.54b, National Institutes of Health, Bethesda, MD, USA) in 5 different areas per picture to ensure representativeness.

### 2.11. Morphological Analysis

Images were obtained using an optical microscope. Cell surface area was calculated from at least 100 cells per condition after calibrating the scale in ImageJ Software to guarantee accurate measurement.

### 2.12. Immunofluorescence Analysis of Epidermal Differentiation on Skin Equivalents

Sections with 5 µm thickness were prepared and subjected to a rehydration protocol. For antigen retrieval, the sections were treated with 0.01 M sodium citrate buffer at pH 6. After antigen retrieval, non-specific binding sites were blocked with 10% goat serum. This was followed by overnight incubation at 4 °C with primary antibodies targeting cytokeratin 10 (Abcam, Waltham, MA, USA, #ab9026) and loricrin (Abcam, Waltham, MA, USA, #ab85679). Subsequently, sections were incubated with appropriate secondary antibodies for 1 h at room temperature. Slides were then mounted using DAKO mounting medium (Agilent, Waldbronn, Germany, S3023) and examined under a Cell Voyager CV1000 Yokogawa confocal microscope (Visitron Systems, Puchheim, Germany).

### 2.13. Bioinformatics

We analyzed data from three prior research projects to identify novel senescence-related genes. Specifically, this analysis incorporated findings from a microarray study conducted on UVB-irradiated HDFs on days 7 and 9 [16], an RNA-seq study performed on UVB-irradiated HDFs on day 4 (unpublished), and an RNA-seq study conducted on tBHP-treated HDFs on day 9 [17]. Using the STAR aligner v.2.7.1a, reads were aligned to the GRCh38/hg38 genome assembly. The NCBI Refseq gene annotation and the featureCounts v2.0.0 [20] tools were used to determine read counts at the gene level. Secondly, we gathered transcriptome information from human fibroblast cells both before and after tBHP treatment [17]. For each gene in the study, logFC values were gathered. Finally, we collected array data from human diploid cells that underwent UVB-induced senescence [16]. The supplemental data were used to determine the logFC values. Only genes that were shared by all platforms were preserved, and those genes were sorted by logFC value. We gathered the top 500 overexpressed genes for each condition to identify the major regulator shared by all models. We kept genes common to each of the Top 500. After filtering, we isolated five candidates for qPCR verification. Heatmaps and Venn diagrams were generated using the R package ComplexHeatmap [21] and the VennDiagram [22] with the log2 fold changes.

### 2.14. Statistics

Unless differently stated, all experiments were performed with at least 3 biological replicates. Results are shown as the means of independent experiments plus standard deviation. To compare differences between conditions, Student’s t test was used. In all graphics, * *p* < 0.5, ** *p* < 0.1, and *** *p* < 0.01.

## 3. Results

### 3.1. CLCA2 Is Upregulated during UVB-Induced Senescence in HDFs

We initially performed an analysis of microarray and RNA-seq data derived from previous research projects conducted within our laboratory. These datasets encompassed various time points from both UVB- and tBHP-induced senescence models, including a microarray study performed on UVB-irradiated HDFs on days 7 and 9 [16], an RNA-seq study conducted on UVB-irradiated HDFs on day 4 (unpublished), and an RNA-seq study carried out on tBHP-treated HDFs on day 9 [17]. The primary aim of this analysis was to identify genes consistently exhibiting upregulation in different models of SIPS. The analysis of these datasets identified five genes, *SERPINB2*, *SCG5*, *WDR63*, *NEFL*, and *CLCA2*, that were consistently upregulated across all datasets (Figure 1A, Appendix A). The upregulation of each candidate was evaluated by RT-qPCR (Appendix A–F), validating the transcript level increase for all candidates during SIPS. *CLCA2* emerged as the most significantly upregulated gene among all candidates, leading to its selection for subsequent experiments and in-depth analysis.

The regulation of CLCA2 protein expression in response to UVB irradiation was further studied by Western blot (Figure 1B,C). The existing literature indicates a complex and not fully understood pattern of posttranslational modifications [23], where the precursor native CLCA2 protein (105 kDa) undergoes processing within the Endoplasmic Reticulum (ER) (128 kDa) and the Golgi apparatus followed by translocation to the plasma membrane (141 kDa). At the plasma membrane, CLCA2 can be cleaved, resulting in shedding of the ectodomain and a small membrane-resident fragment of 35 kDa [23]. Upon UVB treatment (day 4 and day 9), a consistent upregulation of bands around 130/140 kDa was observed. Notably, no changes were observed in the expression of the precursor native protein (105 kDa). In contrast, while we observed an upregulation of CLCA2 at the transcriptomic level in the tBHP model (Appendix A), this increase was not reflected at the protein level (Appendix A).

### 3.2. CLCA2 Is Upregulated during Nutlin-Induced Senescence in HDFs

To further investigate CLCA2-mediated cellular processes and mechanisms underlying CLCA2-associated senescence, we employed the Nutlin3a-induced senescence model for our study. This model was chosen based on the established dependency of CLCA2 on p53 [14]. Nutlin3a (N3) is a small molecule that inhibits the interaction between MDM2 and p53, thus preventing the ubiquitination and degradation of p53 and leading to its stabilization and activity [18]. Consistent with our findings in the UVB model, N3 treatment induced an upregulation of CLCA2 bands in the range of 130–140 kDa (Figure 1D). In addition, we observed the appearance of a CLCA2 band at 35 kDa. While the identity of this band has not been formally determined, the literature suggests that it might be associated with the cleaved form of the plasma membrane protein that persists in the membrane following the release of the ectodomain.

Considering the prediction of CLCA2 localization to the plasma membrane, we aimed to analyze its presence on the cell surface. Flow cytometry analysis revealed that in N3-induced senescent HDFs, CLCA2 was indeed prominently located on the plasma membrane (Figure 1E). In the UVB model, we observed a significant increase in CLCA2 surface expression (Figure 1F), although the effect was not as pronounced as in the N3 model, paralleling the observations from Western blot analysis. Conversely, in the tBHP model where the regulation of CLCA2 protein expression was not observed, its presence on the cell surface was not detected (Appendix A).

### 3.3. CLCA2 Downregulation Promotes Replicative Senescence

To address the role of CLCA2 in senescence, we employed HDFs transduced with lentiviral particles to silence CLCA2 expression (hereafter named CLCA2KD HDFs). The decreased expression of CLCA2 mRNA and protein after transduction was confirmed by RT-qPCR (Appendix A) and Western blot, respectively (Appendix A). CLCA2KD and SCR HDFs were maintained in culture for several passages and monitored for senescence-related parameters in three distinct phases: early passage (EP) (from P10 to P13), middle passage (MP) (from P18 to P21), and late passage (LP) (from P26 to P29). Notably, CLCA2KD HDFs exhibited a slower growth rate compared to the SCR cells from EP to LP (Figure 2A).

The observed reduction in the growth rate of CLCA2KD HDFs coincided with an increase in the number of senescent cells compared to SCR cells, as displayed by the increased percentage of SA-β-gal-positive cells (Figure 2B, Appendix A) and increased cell area (Appendix A), indicative of a more flattened morphology, a hallmark of senescence.

Furthermore, when compared to SCR fibroblasts, CLCA2KD cells displayed decreased levels of the phosphorylated form of retinoblastoma protein (pRb) and of Lamin B1, particularly during EP and MP (Figure 2C–E), suggesting that the depletion of CLCA2 impairs cell cycle progression and induces premature senescence. Furthermore, it is noteworthy that during late passage, SCR HDFs already exhibit typical characteristics of senescent cells (i.e., replicative senescence due to telomere attrition), which may explain why the differences were not as pronounced at this stage.

When investigating the secretome of CLCA2-depleted HDFs, notable differences were observed in the composition of secreted soluble factors compared with SCR HDFs (Figure 2F), particularly during LP. At this stage, the levels of several interleukins and TGF-β1 were notably elevated in CLCA2KD HDFs. Since extracellular vesicles (EVs) are increasingly considered important components of the SASP, we assessed the release of EVs in response to CLCA2KD. The release of EVs was greatly increased in CLCA2KD HDFs in MP and LP (Figure 2G). No significant differences in the size of EVs released by SCR and CLCA2KD HDFs from the same passage were observed (Figure 2H). To unequivocally establish the identity of particles in the conditioned media as EVs, we examined the presence of EV surface markers, such as the tetraspanins CD9, CD63, and CD81 using flow cytometry (FACS) and Imagestream. We found that the majority of EVs were enriched in CD81 (Figure 2I, Appendix A). Altogether, our data suggest that the depletion of CLCA2 induces premature senescence in HDFs characterized by a distinct secretome, including extracellular vesicles.

### 3.4. SEs Prepared with CLCA2 KD HSDFs Mimic Skin Aging Features

To explore the role of CLCA2 in the context of skin aging, we employed 3D skin equivalent (SE) models. These SEs were constructed using patient-matched human skin dermal fibroblasts (HSDFs) and human skin epidermal keratinocytes (HSEKs). The dermal compartments of the skin equivalents (SEs) were generated using human skin dermal fibroblasts (HSDFs) subjected to four different conditions: wild type (WT) untreated, WT UVB day 4 irradiated, SCR untreated, or CLCA2KD untreated. The epidermal compartment consisted of WT untreated HSEKs (Figure 3A). To assess the potential role of CLCA2 in skin homeostasis, we compared SEs prepared with HSDFs exhibiting the upregulation of CLCA2 (UVB-irradiated, Appendix A) with SEs prepared with CLCA2-depleted HSDFs.

SEs prepared with UVB-irradiated senescent HSDFs displayed characteristics of aged skin [24], including a reduction in epidermal thickness and the presence of keratinocytes with apoptotic-like features (Figure 3B). Similarly, SEs prepared with CLCA2KD HSDFs also exhibited decreased epidermal thickness (Figure 3B). Furthermore, the CLCA2KD SE displayed signs of impaired terminal epidermal differentiation, evidenced by a stratum corneum that appeared detached from the underlying epidermal layers. In contrast, the control SE, including WT untreated and SCR SE, displayed a fully differentiated epidermis. No notable changes were observed in the dermal layer across any of the SE models.

To gain deeper insights into the mechanisms underlying impaired epidermal differentiation in response to various types of HSDF senescence, we utilized immunofluorescence (IF) to label the epidermal differentiation markers cytokeratin 10 and loricrin in sections of the four different types of SE. In SEs prepared with WT HSDFs (both treated and untreated) and SCR HSDFs, cytokeratin 10 was expressed in the stratum spinosum, and loricrin was observed at the boundary between the stratum granulosum and stratum corneum (Figure 3C, Appendix A), consistent with their expected layers. In contrast, in CLCA2KD SEs, the expression of cytokeratin 10 and loricrin was predominantly localized in the upper layers of the stratum corneum, where nucleated cells were also observed. Altogether, these observations highlight a potential role for CLCA2 in regulating epidermal homeostasis and suggest that environmental stressors such as UVB may induce CLCA2 expression to safeguard skin homeostasis.

## 4. Discussion

This study explored potential roles of CLCA2 in different models of cellular senescence and demonstrated that CLCA2 expression can be modulated by different senescence triggers such as UVB and Nutlin3a. The knockdown of CLCA2 led to the decreased proliferation and increased expression of senescence markers, indicating a possible role in preventing premature senescence. Furthermore, analysis of the secretome and extracellular vesicle release in CLCA2-depleted cells further implicated CLCA2 in intercellular communication during senescence. In 3D skin equivalent models, the depletion of CLCA2 in dermal fibroblasts resulted in features resembling aged skin, suggesting its involvement in regulating epidermal homeostasis. These findings highlight the significance of CLCA2 in cellular senescence and skin aging and provide insights into its molecular functions and impact on cell and tissue homeostasis.

While the mRNA expression of CLCA2 was upregulated in the all three senescence models studied in this work, the upregulation of CLCA2 protein levels was not observed in the tBHP-induced senescence model of HDFs. This observation highlights the complexity of cellular responses to different stressors. It is known that various types of stressors converge in generalized DNA damage, and the activation of the DNA damage response (DDR) is a key factor in the establishment of different types of senescence [10]. Central to coordinating DDR mechanisms, p53 plays an essential role in cellular senescence. We have previously shown that in UVB- [16] and tBHP-induced senescence [17] models, p53 undergoes stabilization through phosphorylation at Serine 15. Additionally, the stabilization of p53 by the small molecule Nutlin3a is sufficient to induce non-physiological drug-induced senescence [18], highlighting the importance of p53 in senescence induction. It is known that p53 stabilization can occur via various mechanisms, with numerous signaling pathways contributing to the phosphorylation, acetylation, methylation, and ubiquitination of this protein. These post-translational modifications are highly specific to different types of genotoxic stress [25], indicating a complex regulation of p53 activity. The different mechanisms and signaling pathways that are involved in UVB-mediated SIPS, which is known to cause direct DNA lesions like cyclobutane pyrimidine dimers [26] and tBHP-mediated SIPS, which is driven by undermining the antioxidative capacity of HDFs, may result in different p53 stabilization routes, which could explain why CLCA2 protein is upregulated in the UVB-induced senescent HDFs but not in the tBHP model. To further elucidate the role of CLCA2 in cellular senescence, more work will be required to explore its upregulation on mRNA vs. protein level across additional senescence models. Additionally, to unravel the physiological relevance of CLCA2, studies with additional exposome factors may be helpful, like skin aging models combining UV exposure and pollution [27].

During cellular senescence, there is an increase in intracellular calcium levels, which appear to play a pivotal role as an instrumental effector in the process [28]. This calcium overload leads to an increase in mitochondrial calcium, resulting in a decrease in mitochondrial membrane potential and an escalation in the production of reactive oxygen species (ROS) [28]. While the influence of calcium during cellular senescence has been extensively studied, the role of other ions such as chloride has not received as much attention. Nonetheless, chloride is recognized for its significance in cellular physiology, particularly for maintaining osmotic pressure and acid–base regulation [29]. CLCA2 has been reported to influence intracellular calcium levels [30] as well as to bind and regulate chloride channels [12,30], which could be essential as a mechanism to maintain cellular homeostasis in response to different stressors. In this study, we demonstrated that CLCA2 is elevated during different types of cellular senescence and that reducing its expression leads to an early initiation of cellular senescence. Consequently, we propose that CLCA2 plays a protective role and supports cellular homeostasis, as evidenced by its increased expression during UVB- and N3-induced senescence, potentially acting to mitigate cellular stress. The presence of CLCA2 likely helps in slowing down the senescence process, reinforcing our theory that the observed rise in CLCA2 levels functions as a defense mechanism, complementing other protective responses such as enhanced autophagy or DNA damage repair.

We observed that the depletion of CLCA2 in HDFs induces premature senescence with classic senescence markers, yet these cells also displayed a unique SASP composition. The secretome of senescent cells, which varies significantly based on the senescence-inducing stimulus, contributes to the diversity of senescence phenotypes [31]. It is well established that the secretome influences mechanisms of intercellular communication, leading to morphological and functional alterations in tissues. In fact, in the context of the skin, senescent fibroblasts can induce changes in the epidermal compartment leading to the appearance of skin aging characteristics such as decreased epidermal thickness in senoskin models [24]. It is established that the interplay between fibroblasts and keratinocytes through the secretome is crucial for maintaining skin homeostasis [32]. The observation of elevated levels of cytokines, specifically IL-1α and IL-1β, in CLCA2KD HDFs suggests a potential link to increased inflammation [33]. This inflammatory response is further complemented by the elevated levels of TGF-β1, indicating potential alterations in cellular signaling pathways [34]. These alterations could affect tissue repair and fibrosis, as well as possibly play a role in epidermal differentiation.

Epidermal differentiation is a complex process that is essential to the skin’s continuous regeneration. To better understand the implications of CLCA2 during skin development and aging, we developed 3D SE models in which we explored two types of senoskin, the UVB SE and the CLCA2KD SE. Although both models exhibit skin aging phenotypes, notable differences were observed between them. One SE model contained UVB-induced senescent fibroblasts, which upregulate CLCA2 levels and are characterized by alterations in mechanisms of protein quality control and increased levels of intracellular ROS [19,35]. In the other model, the absence of CLCA2 led to the emergence of senescent fibroblasts, characterized by a specific SASP composition and an increased release of EVs. Skin equivalents prepared with HDFs from both models were characterized by a decrease in epidermal thickness, a major characteristic of skin aging. However, the CLCA2KD SE displayed a phenotype of compromised epidermal differentiation, marked by a detached stratum corneum containing nucleated cells. The persistence of nucleated keratinocytes in the stratum corneum is known as parakeratosis, and it is associated with a variety of skin disorders [36]. Thus, the presence of CLCA2 in the fibroblasts of SE appears to act as a protective factor, enabling the proper terminal differentiation of the epidermis. It is possible, therefore, that the ectodomain of CLCA2 might be facilitating this effect on epidermal terminal differentiation. Additionally, CLCA2 might also regulate skin homeostasis by affecting the influx of calcium and chloride ions in epidermal keratinocytes. For instance, skin barrier repair can be accelerated by increased intracellular chloride in keratinocytes which induces the exocytosis of lipid-containing lamellar bodies [37]. Therefore, CLCA2, by regulating chloride channels in the skin, likely plays an important role in maintaining skin homeostasis. Furthermore, CLCA2 seems to preserve skin homeostasis by protecting keratinocytes from hyperosmotic stress and promoting cell–cell adhesion in the epidermis, overall preserving the epidermal barrier function [38].

The relevance of our findings on CLCA2’s function in cellular senescence and skin homeostasis extends to aging interventions and therapies, presenting CLCA2 as a potential marker for identifying senescent cells within photoaged skin. This prospect opens avenues for future research to clarify whether CLCA2’s presence is indicative of various types of cellular senescence in and beyond the skin, including the potential detectability in patient serum. Moreover, CLCA2 emerges as a promising target for anti-aging therapies, underscoring the need for exploration into its therapeutic potential.

In conclusion, our findings indicate that CLCA2 is crucial in the context of UVB- and N3-induced senescence, with its deficiency accelerating the senescence process, evidenced by an increased expression of cytokine and EV release. The SE developed with CLCA2KD resembled characteristics of skin aging and defective epidermal differentiation, highlighting the significance of CLCA2 in the regulation of epidermal homeostasis..

## Figures and Tables

**Figure 1 biomedicines-12-00592-f001:**
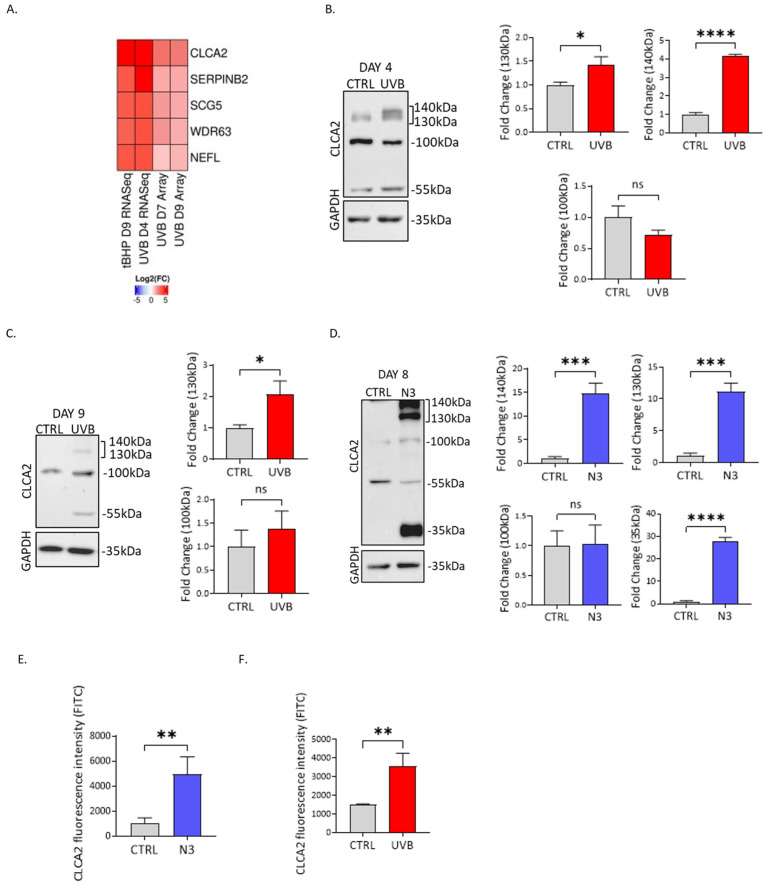
CLCA2 is upregulated in UVB- and Nutlin3a-induced senescence. (**A**) Common genes from 4 data sets (tBHP Day 9 RNAseq, UVB Day 4 RNAseq, UVB Day 7 Array, UVB Day 9 Array) are shown as a heatmap. (**B**) CLCA2 Western blot protein expression in UVB model Day 4. (**C**) CLCA2 Western blot protein expression in UVB model Day 9. (**D**) CLCA2 Western blot protein expression in Nutlin3a (N3) model Day 8. (**E**) CLCA2 fluorescence intensity from FACS N3-induced senescent HDFs (Day 8). (**F**) CLCA2 fluorescence intensity from surface from FACS UVB-induced senescent HDFs (Day 9). The 55kDa band from Western blot images was not previously described as a CLCA2 polypeptide. Data represent mean values ± SD, N = 3. In all graphics, ns: non-significant; * *p*  <  0.05; ** *p*  <  0.01; *** *p*  <  0.001; **** *p*  <  0.0001.

**Figure 2 biomedicines-12-00592-f002:**
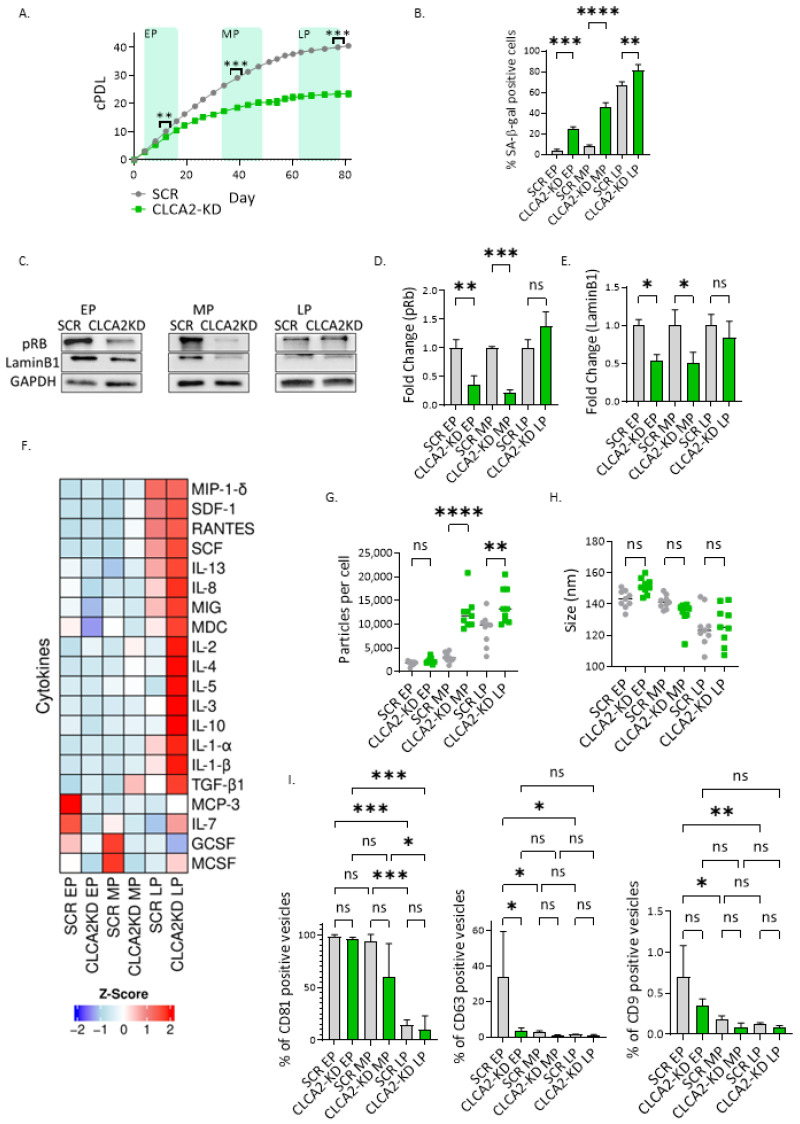
CLCA2 knockdown impedes cell proliferation in HDFs showing senescence features. (**A**) Growth curve showing cumulative population doublings (cPDL) in SCR and CLCA2KD HDFs. (**B**) Quantification of SA-β-galactosidase-positive cells in SCR and CLCA2KD HDFs. (**C**) Phosphorylated retinoblastoma protein and LaminB1 Western blot protein expression in SCR and CLCA2KD HDFs. (**D**) pRB densitometric analysis. (**E**) LaminB1 densitometric analysis. (**F**) Z-score normalized data from cytokine array from SCR and CLCA2KD at different passages. (**G**) Particles per cell. (**H**) Size from isolated EVs from SCR and CLCA2KD HDFs. (**I**) Characterization of the percentage of CD81-, CD63-, and CD9-positive particles in SCR and CLCA2KD HDFs. Data represent mean values ± SD, N = 3, except from cytokine array, where N = 2. In all graphics, ns: non-significant; * *p*  <  0.05; ** *p*  <  0.01; *** *p*  <  0.001; **** *p*  <  0.0001.

**Figure 3 biomedicines-12-00592-f003:**
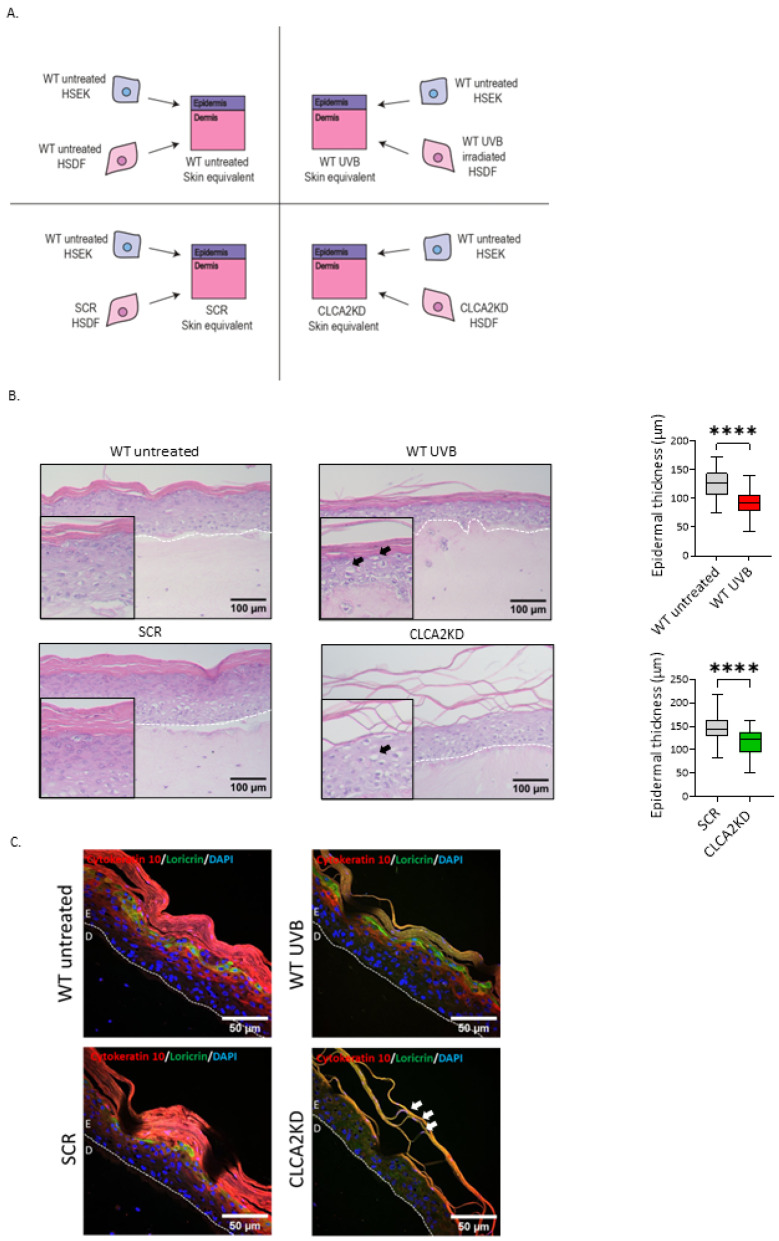
CLCA2KD SEs show impaired epidermal differentiation. (**A**) Scheme of SE production. (**B**) H-E representative pictures and epidermal thickness measurements from untreated WT, UVB WT, SCR, and CLCA2KD SEs. Black arrows show keratinocytes with apoptotic-like features. (**C**) Cytokeratin 10 and loricrin immunofluorescence pictures from untreated WT, UVB WT, SCR, and CLCA2KD SEs. White arrows show nucleated cells. Data represent mean values ± SD, N = 3. In all graphics, ns: non-significant; **** *p*  <  0.0001.

## Data Availability

Data are available upon reasonable request.

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
