# Peer review of "CLCA2: A Potential Guardian against Premature Senescence and Skin Aging"

_biomedicines, 2024, doi:10.3390/biomedicines12030592_

Round 1

Reviewer 1 Report

Comments and Suggestions for Authors

In the present manuscript, it indicated that the CLCA2, a calcium-activated chloride channel accessory protein, was upregulated in UVB- and Nutlin3a-induced senescence models at both transcriptomic and proteomic level. In 3D skin equivalent model, CLCA2 knockdown fibroblasts exhibited the feature of the aged skin. These results suggested that CLCA2 as a novel regulator of cellular senescence. However, the manuscript is not well-written. I recommend that this paper accepted after minor revision.

1. It is better to show the result of the CLCA2 knockdown in UVB- and Nutlin3a-induced senescence models. If the CLCA2 is a regulator of cellular senescence, the CLCA2 knockdown in the senescence models predicts the accelerate aging features.

2. Is the expression of the CLCA2 upregulating in the actual fully aged cells? Or is the expression of the CLCA2 already decreased because the CLCA2 could not suppress aging?

3. It is difficult for me to understand the result in Figure 2I. Does the result indicate that the CLCA2 depletion induce the senescence in HDF? What is the premature senescence?

4. The index is not on top in line 97, 102, 106, 143 and 190.

Author Response

Answer to reviewers’ comments

Reviewer 1

In the present manuscript, it indicated that the CLCA2, a calcium-activated chloride channel accessory protein, was upregulated in UVB- and Nutlin3a-induced senescence models at both transcriptomic and proteomic level. In 3D skin equivalent model, CLCA2 knockdown fibroblasts exhibited the feature of the aged skin. These results suggested that CLCA2 as a novel regulator of cellular senescence. However, the manuscript is not well-written. I recommend that this paper accepted after minor revision.

  1. It is better to show the result of the CLCA2 knockdown in UVB- and Nutlin3a-induced senescence models. If the CLCA2 is a regulator of cellular senescence, the CLCA2 knockdown in the senescence models predicts the accelerate aging features.

Reply: In our study we provided evidence that depletion of CLCA2 leads to accelerated senescence, suggesting that the presence of CLCA2 serves to delay the senescence process. This observation supports our hypothesis that the upregulation of CLCA2 observed in both UVB and Nutlin3a models acts as a protective mechanism, similar to other protective measures like increased autophagy or DNA damage response, which are recognized in the field (doi.org/10.3390/antiox12010169; doi:10.1038/nrc2440).Taking into account the protective role of CLCA2, we predict that subjecting CLCA2-depleted cells to UVB and Nutlin3a-induced senescence models would either accelerate senescence or predispose cells to apoptosis. This hypothesis aligns with our previous findings, where blocking a protective mechanism such as autophagy through the suppression of Atg7 alters the fate of HDFs, shifting them from senescence to apoptosis when subjected to UVB irradiation (Cavinato, 2016, doi.org/10.1093/gerona/glw150).

In response to the reviewer's comment we have rephrased the paragraph in the discussion section of the revised manuscript (lines 456-459) to better explain this specific hypothesis.

  1. Is the expression of the CLCA2 upregulating in the actual fully aged cells? Or is the expression of the CLCA2 already decreased because the CLCA2 could not suppress aging?

Reply: Thank you for raising this important question concerning the expression dynamics of CLCA2 in fully aged cells. Our findings, as illustrated in Figures 1C and 1D, demonstrate that CLCA2 expression is indeed upregulated in cells that have undergone senescence induced by UVB irradiation and Nutlin3a treatment. This upregulation suggests that CLCA2 functions as a protective factor in the context of cellular senescence, a hypothesis supported by our observations that the absence of CLCA2 accelerates the onset of senescence in a model of replicative senescence. When cells are exposed to harmful conditions, such as UVB light, they respond by activating defense mechanisms like autophagy and DNA repair, which help lessen the damage and keep the cells alive. However, if the damage is excessive, the cells, despite their efforts to enhance these protective mechanisms, ultimately reach a state of senescence. Our research indicates that CLCA2 is a component of the cell's defense mechanisms, helping the cell to survive under stress. In response to the reviewer's comment, we have rephrased the paragraph in the discussion section of the revised manuscript (lines 456-459) to better explain this specific hypothesis.

  1. It is difficult for me to understand the result in Figure 2I.

Reply: Thank you for your inquiry regarding Figure 2I and its implications for our study on CLCA2 depletion in HDF. To clarify, Figure 2I presents the analysis of tetraspanin profiles within the extracellular vesicles (EVs) as a method to authenticate the presence of bona fide EVs. This analysis was part of a broader effort to explore the SASP composition in cells subjected to CLCA2KD compared to SCR cells. In response to the reviewer's comment, we have added a sentence in the results section (lines 341-348) of the revised manuscript to better explain the data shown in Fig. 2I

Does the result indicate that the CLCA2 depletion induce the senescence in HDF? What is the premature senescence?

Reply: The induction of premature senescence, as observed in our study, refers to the accelerated entry into the senescent state (replicative senescence) of HDF cells following CLCA2 depletion. The results revealed that CLCA2KD notably accelerates the onset of replicative senescence. This was shown by a significant reduction in cell proliferation rates and the appearance of senescence markers, such as elevated β-galactosidase activity or decreased Lamin B1 levels, compared to the SCR cells.

  1. The index is not on top in line 97, 102, 106, 143 and 190.

Reply:

In response to this reviewer comment, we meticulously reviewed the mentioned lines (97, 102, 106, 143, and 190) in the version of the manuscript that was submitted. Initially, we did not identify any issues. However, we acknowledge that formatting issues can sometimes be elusive or vary depending on the software and version used for viewing the document.

Reviewer 2 Report

Comments and Suggestions for Authors

Please address the following. Expand the discussion of potential mechanisms by which CLCA2 regulates senescence and skin homeostasis. Provide more details on the skin equivalent models - age and gender of skin donors, number of donor samples tested. Include demographic/descriptor data for human samples in Table 1. Discuss relevance of findings for aging interventions or therapies.

Author Response

Reviewer 2

Please address the following. Expand the discussion of potential mechanisms by which CLCA2 regulates senescence and skin homeostasis.

Reply:

In response to your valuable feedback, we have expanded our discussion on the potential mechanisms by which CLCA2 regulates skin homeostasis, now detailed in lines 492-500 of the revised manuscript. The discussion is based in the understanding that chloride anions play a critical role in epidermal function, particularly in keratinocytes, where Cl- is essential for preserving the integrity and homeostasis of the skin. The mechanism by which CLCA2 modulates skin homeostasis can be appreciated through its regulatory effect on calcium-activated chloride channels. The influx of chloride is pivotal for several physiological processes in keratinocytes, including the exocytosis of lamellar bodies and the protection from hyperosmotic stress, a condition that can compromise skin integrity and function. (doi.org/10.1046/j.1523-1747.2003.12367.x, doi: 10.1126/scitranslmed.aao4650). In response to the reviewer's comment we have added a sentence to the discussion section of the revised manuscript (lines 492-500) to better explain this specific hypothesis.

Provide more details on the skin equivalent models - age and gender of skin donors, number of donor samples tested. Include demographic/descriptor data for human samples in Table 1.

Reply:

We appreciate your request for additional details regarding the skin donor samples used in our skin equivalent models. The skin equivalents were constructed using Human Skin Dermal Fibroblasts (HSDF) and Human Skin Epidermal Keratinocytes (HSEK) derived from a single donor, a 35-year-old male. Experiments were conducted in technical triplicates to ensure reproducibility and reliability of the results. Due to the ethical guidelines and confidentiality agreements with the Department of Plastic, Reconstructive, and Aesthetic Surgery at the Medical University of Innsbruck, we are limited in the amount of personal information we can obtain about the donors. This protocol is in place to protect donor anonymity and ensure compliance with privacy regulations. Consequently, the specific demographic and descriptor data for the human samples are not provided to our research team.

We have added a clarification in the methods section (lines 201-202) to offer further details on the skin donor and the production of skin equivalents.

Discuss relevance of findings for aging interventions or therapies.

Reply:

We appreciate your suggestion to elaborate on the potential implications of our findings for aging interventions and therapies. In response, we have expanded our discussion in the manuscript (lines 501-507) to better underscore the significance of CLCA2 in the context of cellular senescence and skin homeostasis, with a particular focus on its applicability to aging-related therapeutic strategies.